# Treatment Outcomes in Spinal Tumors According to Patients’ Perspectives: A Focus on Indeterminate Spinal Instability

**DOI:** 10.3390/curroncol32010038

**Published:** 2025-01-13

**Authors:** Victoria H. Schimmelpenning, Robin Brugger, Nikki Rommers, Johann Kunst, Martin Jäger, Christoph E. Albers, Helena Milavec

**Affiliations:** 1Spine Unit, Department of Orthopaedic Surgery and Traumatology, Aarau Cantonal Hospital, 5001 Aarau, Switzerland; brugger@schmerzzentrum.ch (R.B.); johann.kunst@ksa.ch (J.K.); martin.jaeger@ksa.ch (M.J.); 2Department of Clinical Research, University of Basel, University Hospital Basel, 4031 Basel, Switzerland; nikki.rommers@usb.ch; 3Department of Orthopaedic Surgery and Traumatology, University Hospital Bern, Inselspital, University of Bern, 3010 Bern, Switzerland; christoph.albers@insel.ch

**Keywords:** spinal neoplastic lesions, indeterminate spinal instability, spinal instability neoplastic score (SINS), tumor, treatment outcomes

## Abstract

The objective of this study was to analyze treatment approaches and outcomes according to patients’ perspectives for patients with indeterminate spinal instability caused by neoplastic lesions. Data were collected from 31 patients with a total of 147 spinal neoplastic lesions, 29 of whom had lesions classified as indeterminate. These lesions were divided into two groups: the low indeterminate group (SINS 7–9) and the high indeterminate group (SINS 10–12). Conservative treatment was the primary approach (93%), resulting in improvement in 59% of cases, stability in 22%, and asymptomatic outcomes in 19%. No significant differences in self-reported outcomes were found between surgical and non-surgical treatments (*p* = 0.98, *p* = 0.18). Surgery was reserved for patients with severe pain or impending neurological compromise. Our findings suggest that conservative management is a viable option for most patients with indeterminate spinal instability caused by neoplastic lesions, provided pain and neurological stability are adequately controlled.

## 1. Introduction

With rising global cancer rates, spinal neoplastic lesions are becoming more common, affecting up to 36% of individuals with malignancies, particularly those with breast, lung, and prostate cancers. Among primary neoplasms, multiple myeloma is the most common condition affecting the spine. Malignant spinal involvement poses a significant threat, often leading to spinal instability, pathological vertebral fractures, or metastatic epidural spinal cord compression, resulting in increased pain and reduced quality of life for these cancer patients [1,2,3,4,5,6].

Assessing spinal neoplastic conditions involves evaluating tumor-induced spinal instability, typically determined using the spinal instability neoplastic score (SINS). SINSs ranging from 0 to 6 indicate biomechanical stability, while scores of 13 to 18 indicate instability and a high fracture risk. The indeterminate range, which suggests a possibly impending risk, falls between 7 and 12. The SINS serves as a valuable tool to assist clinicians in making informed treatment decisions [7].

Treatment options for spinal neoplastic lesions are diverse and depend on critical factors such as age, overall health, comorbidities, life expectancy, disease extent, and lesion stability. These options include non-surgical interventions such as pain management, physiotherapy, and radiotherapy [8]. For cases requiring surgical intervention, techniques such as percutaneous vertebral augmentation and the use of spinal implants, including radiolucent carbon-fiber implants, are commonly employed to restore spinal stability [9]. Currently, treatment options for patients with indeterminate spinal instability facing an impending fracture risk within the SINS 7–12 range are under ongoing evaluation [10,11,12,13]. Several approaches have been proposed to determine a cutoff score within this gray zone, with Vargas et al. suggesting a threshold of 11 and Mahakul et al. recommending a score of 9 [14,15].

Given the absence of clear thresholds in this gray zone, our effort was to emphasize patients’ self-reported experiences over the following months as a key factor in evaluating outcomes in patients with indeterminate spinal instability, rather than relying solely on radiological findings. The aim of this study was to provide an overview of treatments and treatment outcomes according to patients’ perspectives, with a specific focus on those with indeterminate spinal instability. We further stratified the indeterminate group into two subgroups: those with a lower indeterminate SINS (7–9) and those with a higher indeterminate SINS (10–12).

## 2. Material and Methods

This retrospective study utilized data collected from patients diagnosed with spinal neoplastic lesions at our tertiary institution. The study population consisted of patients referred to the spinal unit in the orthopedic department who provided written informed consent in accordance with the hospital’s general guidelines. Ethics approval for this study was obtained from the relevant ethics board prior to commencement.

### 2.1. Data Collection

Electronic medical records of patients attending our outpatient clinic with symptomatic or asymptomatic malignant spinal involvement between January 2021 and May 2023 were reviewed. The number of visible neoplastic lesions per patient varied widely. In cases with multiple lesions, we recorded the most severe lesions based on their SINS, up to a maximum of 10 lesions per patient.

Demographic details such as age, sex, and primary cancer type, along with SINS and tumor extent information (Bilsky score/Enneking spinal column classification), were collected from the medical records. Additionally, data on treatments (e.g., radiotherapy, chemotherapy, surgical technique) were extracted. Each parameter of the SINS, including location, mechanical back pain, type of bone lesion, radiographic spinal alignment, vertebral body collapse, and posterolateral involvement, was examined. The analysis specifically focused on lesions in the indeterminate fracture risk subgroup (SINS 7–12).

To gain further insight into the broader group of patients with indeterminate fracture risk, we stratified them into two subgroups: the low indeterminate group (SINS 7–9) and the high indeterminate group (SINS 10–12).

### 2.2. Pain Evaluation

For pain evaluation, we routinely used the numeric rating scale (NRS) or visual analogue scale (VAS) during each consultation. Patients are asked to rate their pain, and if the reported score is lower, it is recorded as “improved/better”. If the score remains the same, it is noted as “equal,” and if the score is higher, it is recorded as “worse”. This standardized approach ensured consistent and reliable tracking of pain levels throughout the study.

### 2.3. Inclusion and Exclusion Criteria

#### 2.3.1. Inclusion Criteria

All patients aged 18 years or older who underwent at least one follow-up visit after treatment for spinal neoplastic lesions were considered eligible for inclusion in the study. We included patients only if both a standing X-ray and an MRI were available for review. Additionally, all patients had provided written informed consent, ensuring their voluntary participation in the study. Only patients with complete baseline clinical, radiological, and demographic data available for analysis were included, allowing for a comprehensive evaluation of treatment outcomes. Only patients without any relevant communication barriers were included to ensure effective communication.

#### 2.3.2. Exclusion Criteria

Patients were excluded if they met any of the following conditions: under 18 years of age, incomplete baseline clinical, radiological, or demographic data, no follow-up visit after treatment, absence of both standing X-ray and MRI, language barriers that impeded communication or informed consent, and pregnancy.

### 2.4. Primary Outcome

Treatment outcomes were assessed at the lesion level, with patients asked about pain in each spinal region. The primary outcome was measured as closely as possible within 90 days after treatment. Each individual lesion’s treatment outcome was evaluated based on reported pain during patient visits. Treatment outcomes were categorized as follows:Better;Equal;Worse;Never experienced pain/remained pain-free.

For analysis, treatment outcomes were dichotomized into “better” versus “equal or worse”.

### 2.5. Statistics

All analyses were performed using R version 4.3.1 (R Core Team, 2023). Patient characteristics, primary tumor characteristics, and the number of spinal neoplastic lesions are presented as means and standard deviations, medians and interquartile ranges (IQR), or frequency and proportion, as appropriate. Additionally, scores of different components of the SINS were described.

The success of the treatment was evaluated individually for metastasis and summarized by SINS category. The association between treatment and outcome was assessed via mixed-effect logistic regression models, including “patient” as a random intercept to account for the correlation between multiple lesions within a single patient. We present the odds ratio (OR) with 95% confidence interval (CI) for the fixed effect of treatment (surgery (yes over no) or radiotherapy (yes over no).

Patient survival is presented as the number and percentage of patients who died before data extraction and the median survival time after the date of first contact. The proportion of patients who died was stratified by treatment (surgery vs. no surgery), and a Kaplan–Meier curve was plotted to visualize survival by treatment group.

## 3. Results

A total of 31 patients (17 males (54.8%), 14 females (45.2%)) were included in the final analysis, collectively presenting with 147 spinal neoplastic lesions (Appendix A). Patients were followed for a median of 133 days (IQR: 83.75, 283) and attended a median of 2.5 follow-up visits (IQR: 1, 4). The number of lesions per patient ranged from one to multiple, with a median of seven lesions per patient. Most lesions were located in the thoracic and lumbar spine, accounting for 57.4% and 30.7%, respectively. In sum, 44 lesions were categorized as SINS 0–6 (stable), 101 as SINS 7–12 (indeterminate), and 2 as SINS 13–18 (unstable). A total of 29 patients had at least one lesion with indeterminate instability. Table 1 provides an overview of the baseline characteristics of these patients with indeterminate instability, while Table 2 presents details of the spinal neoplastic lesions within the indeterminate group. Table 3 offers a detailed breakdown of each individual lesion, outlining the calculation process and the contribution of each component to the overall score.

### 3.1. Treatment Strategies for the Indeterminate Group

Within the 29 patients in the indeterminate group, most were managed with a non-surgical approach (24 patients, 82.8%). Five patients (17.2%) required surgical intervention, which was performed when pain became unmanageable or when there was an impending risk of neurological impairment. Surgery was recommended for an additional three patients. One patient had disseminated involvement, including four lesions classified with an indeterminate SINS, while another had disseminated lesions, with three categorized as high risk of instability. However, both patients declined surgery and opted for conservative treatment, and one patient could not undergo surgery due to hospital capacity constraints. An overview of patient and tumor characteristics for those treated surgically is provided in Table 4.

Of all patients within the indeterminate group, 23 (79.3%) received radiotherapy and 27 (93.1%) received chemotherapy.

### 3.2. Outcomes in the Indeterminate Group

Within each patient, all lesions showed consistent treatment outcomes. The majority of lesions in both the high and low indeterminate groups either improved (n = 59, 58.4%) or remained pain-free (n = 19, 18.8%). A total of 23 lesions (22.8%) showed no improvement after 90 days, but none of the patients experienced worsening symptoms (Table 5). The treatment outcomes, stratified by treatment type, are displayed in Table 6.

The outcome for most patients receiving conservative treatment was positive. Nineteen patients (65.5%) improved, and eight patients (27.5%) remained pain-free. Similarly, the outcome for most patients undergoing surgery was favorable. Six patients (85.7%) improved, and one patient (14.3%) remained pain-free.

The results of the models showed no significant difference in treatment success between patients treated with and without surgery (OR 1.80 (95%-CI 0.00 to 700.61), *p* = 0.916) or patients treated with and without radiotherapy (OR 1.05 (95%-CI 0.01 to 166.91), *p* = 0.986). Furthermore, within the group with indeterminate SINSs (SINS 7–12), no difference in treatment success was observed between patients treated with or without surgery or radiotherapy (OR 1.89 (95%-CI 0.00 to 68.43), *p* = 0.906; OR 0.91 (95%-CI 0.00 to 223.75), *p* = 0.973, respectively).

### 3.3. Patient Survival

Of the 31 patients, 53.6% died before the date of data extraction. The median survival time from the date of first contact for those who died was 214.5 days (IQR: 118.2, 319.5). Of those treated with surgery, 60.0% died compared to 52.2% of patients treated conservatively. The median survival time was 153 days (IQR: 102, 240) in the surgery group and 223 days (IQR: 131, 317) in the conservative group (Figure 1).

## 4. Discussion

This study provides valuable insights into the management of patients with spinal malignancies facing indeterminate fracture risk (SINS 7–12), a category for which therapeutic recommendations are not yet clearly defined. Among the outpatient population, we found that the majority of patients with indeterminate spinal instability can be managed non-surgically with favorable outcomes according to patients’ perspectives, regardless of whether their SINS falls on the higher or lower end of the indeterminate range.

In palliative care settings, the primary treatment goals are pain reduction, preservation of neurological function and stability, and consequently improved quality of life. In tumor patients, treatment success is primarily based on self-reported outcomes, which are often more relevant than long-term radiological data. These outcomes reflect the patient’s well-being and functional status rather than structural changes.

The primary indications for surgery include instability or neurological impairment [8]. Several scoring systems and algorithms have been developed to assist clinicians in making treatment decisions for patients with spinal neoplastic lesions. The most common score to establish stability is the SINS, a six-item score to define stability in metastatic spinal disease [7]. A systematic review and meta-analysis by Pennington et al. demonstrated good intra- and inter-observer reliability, which improved with greater clinical experience in spinal neoplastic disease [16]. However, 20 years after its introduction by Fisher et al. in 2010, there are still no clear recommendations for further therapy in the broad subgroup of “indeterminate instability” (SINS 7–12) [10,11]. As a result, managing spinal neoplastic lesions in the indeterminate SINS range (7–12) remains challenging, and the limited literature on this topic highlights this issue.

In our population, most patients received chemotherapy, radiotherapy, or a combination of both, along with pain medication and antiresorptive treatment. Surgery was recommended for eight patients with indeterminate risk SINSs, with the primary indication being impending neurological impairment. This aligns with an approach commonly applied to patients with neoplastic lesions and myeloma [6]. Two patients who were recommended for surgery declined the procedure. Notably, both reported pain improvement at the first follow-up, although they experienced persistent pain during subsequent visits to the oncological department. One of these patients passed away two months after presenting to our spine unit.

Few studies have focused on analyzing conservative treatment in spinal tumor patients. A retrospective study by Pennington et al. examined 51 patients with 436 lesions [10]. With findings similar to ours, they concluded that patients with a low indeterminate SINS can be treated conservatively with more confidence, as 80% of their surgically treated patients had a SINS > 10. However, their study did not specifically address the outcomes within the group of indeterminate fracture risk.

Versteeg et al. conducted an international multicenter prospective observational study evaluating data from 307 patients with spinal neoplastic lesions, examining the correlation between SINSs and patient-reported outcome measures (PROMs) [17]. Similarly to our patient cohort, most patients (61.1%) were in the indeterminate SINS group (SINS 7–12). Contrary to our findings, the majority of cases (56.7%) underwent surgical treatment, either alone or in combination with radiotherapy, while the remaining 43.3% were treated solely with radiotherapy. Within the indeterminate group, 67.8% underwent surgery with additional radiotherapy. Their results demonstrated that the total SINS and mechanical pain were significantly, though moderately, associated with pain and physical function.

Another large retrospective study by Vargas et al. examined 75 patients with 292 neoplastic lesions classified as SINS 7–12 [14]. After initial conservative treatment, 34.7% of these patients were transitioned to surgery, primarily due to spinal cord compression, severe pain, or impending cord compression caused by tumor growth. Patients with a higher SINS of 12 were more likely to undergo surgical intervention. They found that specific SINS components, such as lesion type (lytic, mixed, or blastic) and posterior involvement (bilateral, unilateral, or none), were significantly associated with the need for surgery. They concluded that lesions with a SINS > 10 carry a higher risk of instability. In contrast, we could not find any correlation between specific SINS components and the need for surgery within the indeterminate group. A recent study by Mahakul et al. focused on refining the “potentially unstable” spinal instability neoplastic score (SINS) range of 7–12, identifying a SINS value of 9 as a potential cutoff for recommending surgical instrumentation [15].

To summarize, according to patients’ self-reported outcomes, the majority of our patients facing an impending fracture risk due to malignancy either improved or remained stable with conservative treatment. From a patient’s perspective, it may be suggested that non-surgical treatment can be pursued with greater confidence for individuals facing indeterminate spinal instability, given that there was no deterioration in neurological function and pain was manageable. Non-surgically treated patients with higher SINSs within the indeterminate group exhibited similar self-reported outcomes to those with lower SINSs.

### Limitations

Our patient selection was limited to those seen at our outpatient clinics, suggesting a relatively sufficient level of pain control and manageable ambulatory status within this cohort. While this aspect must be acknowledged as a selection bias, it may also serve as a crucial consideration in decision-making.

One limitation of the study is its retrospective design, which may have introduced inconsistencies in data collection due to its nature. Another limitation is the small sample and reliance on data collected from a single tertiary institution, which may limit the generalizability of the findings. The results of this study need to be confirmed in larger, multicenter, ideally prospective studies. Additionally, the variability of primary tumors among the included patients may introduce confounding variables.

It must be noted that there is a dependence of lesions within the same patient (e.g., all lesions within one patient could react similarly to systemic treatment), and that the number of lesions highly differs between patients. This makes the lesion-level results potentially prone to misinterpretation.

## 5. Conclusions

Most patients with impending spinal instability due to neoplastic lesions were treated non-surgically, yielding favorable outcomes according to patients’ perspectives. The majority experienced improvements or remained asymptomatic or stable, underscoring the effectiveness of conservative management in these patients. Non-surgically treated patients with higher SINSs within the indeterminate group exhibited similar outcomes to those with lower SINSs.

According to patients’ self-reported outcomes, conservative treatment for patients with indeterminate spinal instability due to neoplastic lesions was highly effective, with most patients showing improvement or demonstrating no progression of symptoms. Our findings support conservative management as a viable option for this patient population.

## Figures and Tables

**Figure 1 curroncol-32-00038-f001:**
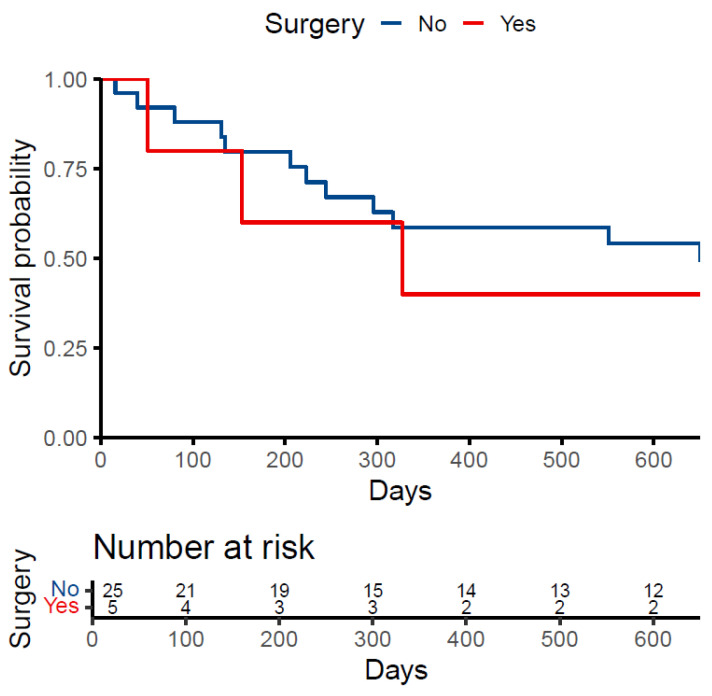
Kaplan-Meier Survival Curve.

**Table 1 curroncol-32-00038-t001:** Patient baseline characteristics.

	Overall (N = 29)
Age (mean (SD))	66.9 (9.8)
Sex = male (n (%))	16 (55.2)
Primary tumor (n (%))	
Lung	7 (24.1)
Lymphoma	1 (3.4)
Breast	4 (13.8)
Multiple Myeloma	7 (24.1)
Myxofibrosarcoma	1 (3.4)
Ovarian	1 (3.4)
Prostate	7 (24.1)
CUP *	1 (3.4)
Outcome (n (%))	
Better	16 (55.2)
Never pain	6 (20.7)
Same	7 (24.1)

*: Cancer of unknown primary origin, SD: standard deviation.

**Table 2 curroncol-32-00038-t002:** Characteristics of the metastases and treatments within the intermediate group.

	Overall	SINS 7–9	SINS 10–12	*p*-Value
n	101	74	27	
X-ray performed (n (%))	96 (95.0)	72 (97.3)	24 (88.9)	0.228
MRI performed (n (%))	94 (93.1)	67 (90.5)	27 (100.0)	0.225
CT performed (n (%))	99 (99.9)	73 (100.0)	26 (100.0)	
Location (n (%))				0.127
cervical	8 (7.9)	5 (6.8)	3 (11.1)	
lumbar	31 (30.7)	24 (32.4)	7 (25.9)	
sacral	4 (4.0)	1 (1.4)	3 (11.1)	
thoracic	58 (57.4)	44 (59.5)	14 (51.9)	
SINS (mean (SD))	8.62 (1.40)	7.95 (0.89)	10.48 (0.64)	<0.001
Radiotherapy (n (%))	55 (54.5)	38 (51.4)	17 (63.0)	0.417
Chemotherapy (n (%))	85 (90.4)	64 (94.1)	21 (80.8)	0.115
Osteometabolic therapy (n (%))	58 (58.0)	45 (61.6)	13 (48.1)	0.324
Surgery (n (%))	7 (8.0)	5 (8.2)	2 (7.7)	1.000

MRI: magnetic resonance imaging, CT: computed tomography, SINS: spinal instability neoplastic score, SD: standard deviation.

**Table 3 curroncol-32-00038-t003:** Description of SINS elements by SINS category.

	Overall	SINS 7–9	SINS 10–12	*p*-Value
n	101	74	27	
Location (n (%))				0.006
Junctional	39 (38.6)	22 (29.7)	17 (63.0)	
Mobile spine	23 (22.8)	21 (28.4)	2 (7.4)	
Semi-rigid	39 (38.6)	31 (41.9)	8 (29.6)	
Mechanical pain (n (%))				0.001
Yes	26 (25.7)	19 (25.7)	7 (25.9)	
No	46 (45.5)	27 (36.5)	19 (70.4)	
Pain free lesion	29 (28.7)	28 (37.8)	1 (3.7)	
Bone lesion (n (%))				0.109
Lytic	64 (63.4)	51 (68.9)	13 (48.1)	
Mixed	36 (35.6)	22 (29.7)	14 (51.9)	
Blastic	1 (1.0)	1 (1.4)	0 (0.0)	
Spinal alignment = Normal (n (%))	55 (55.6)	47 (64.4)	8 (30.8)	0.006
Vertebral body collapse (n (%))				0.006
Collapse with >50% body involved	20 (19.8)	9 (12.2)	11 (40.7)	
Collapse with <50% body involved	27 (26.7)	21 (28.4)	6 (22.2)	
No collapse with >50% body involved	39 (38.6)	34 (45.9)	5 (18.5)	
None	15 (14.9)	10 (13.5)	5 (18.5)	
Posterolateral involvement (n (%))				<0.001
Bilateral	15 (15.2)	5 (6.8)	10 (38.5)	
Unilateral	16 (16.2)	14 (19.2)	2 (7.7)	
None	68 (68.7)	54 (74.0)	14 (53.8)	

SINS: spinal instability neoplastic score.

**Table 4 curroncol-32-00038-t004:** Patient, tumor, and SINS characteristics of the patients who underwent surgery.

	SINS 7–9	SINS 10–12
n	5	2
Age (mean (SD))	67.40 (2.88)	63.00 (2.83)
Sex = male (n (%))	3 (60.0)	1 (50.0)
Primary tumor (n (%))		
Lung	2 (40.0)	1 (50.0)
Lymphoma	1 (20.0)	0 (0.0)
Breast	0 (0.0)	0 (0.0)
Multiple Myeloma	1 (20.0)	0 (0.0)
Myxofibrosarcoma	0 (0.0)	0 (0.0)
Ovarian	0 (0.0)	0 (0.0)
Prostate	1 (20.0)	1 (50.0)
unknown	0 (0.0)	0 (0.0)
X-ray performed (n (%))	5 (100.0)	2 (100.0)
MRI performed (n (%))	5 (100.0)	2 (100.0)
CT performed (n (%))	5 (100.0)	2 (100.0)
Location (n (%))		
cervical	1 (20.0)	0 (0.0)
lumbar	3 (60.0)	0 (0.0)
thoracic	1 (20.0)	2 (100.0)
SINS (mean (SD))	8.40 (0.89)	11.00 (0.00)
Radiotherapy (n (%))	3 (60.0)	1 (50.0)
Chemotherapy (n (%))	5 (100.0)	2 (100.0)
Osteometabolic therapy (n (%))	4 (80.0)	1 (50.0)
Surgery (n (%))	5 (100.0)	2 (100.0)
SINS—Location (n (%))		
Junctional	1 (20.0)	2 (100.0)
Mobile spine	3 (60.0)	0 (0.0)
Semi-rigid	1 (20.0)	0 (0.0)
SINS—Mechanical pain (n (%))		
Yes	2 (40.0)	1 (50.0)
No	1 (20.0)	1 (50.0)
Pain free lesion	2 (40.0)	0 (0.0)
SINS—bone lesion = Mixed (n (%))	3 (60.0)	2 (100.0)
SINS—spinal alignment = Normal (n (%))	3 (60.0)	1 (50.0)
SINS—body collapse (n (%))		
Collapse with >50% body involved	2 (40.0)	1 (50.0)
Collapse with <50% body involved	1 (20.0)	0 (0.0)
No collapse with >50% body involved	1 (20.0)	1 (50.0)
None	1 (20.0)	0 (0.0)
SINS—posterolateral involvement = None (n (%))	4 (100.0)	1 (50.0)
Description surgery (%)		
Dorsal Instrumentation T10–L2	0 (0.0)	1 (50.0)
Dorsal Instrumentation T10–L2	1 (20.0)	0 (0.0)
Corpectomy with dorsal instrumentation	1 (20.0)	0 (0.0)
Kyphoplasty L3	1 (20.0)	0 (0.0)
Kyphoplasty T12–L2	1 (20.0)	0 (0.0)
Operation terminated due to complications	0 (0.0)	1 (50.0)
Kyphoplasty L4	1 (20.0)	0 (0.0)

SD: standard deviation, MRI: magnetic resonance imaging, CT: computed tomography, SINS: spinal instability neoplastic score.

**Table 5 curroncol-32-00038-t005:** Treatment success by SINS-score category.

	Overall	SINS 7–9	SINS 10–12
n	101	74	27
Treatment success (%)			
better	59 (58.4)	42 (56.8)	17 (63.0)
same	19 (18.8)	18 (24.3)	5 (18.5)
worse	0 (0.0)	0 (0.0)	0 (0.0)
never pain	23 (22.8)	14 (18.9)	5 (18.5)

**Table 6 curroncol-32-00038-t006:** Treatment success by treatment type.

	Conservative	Surgery
n	94	7
Treatment success (%)		
better	53 (56.4)	6 (85.7)
same	19 (20.2)	0 (0.0)
worse	0 (0.0)	0 (0.0)
never pain	22 (23.4)	1 (14.3)

## Data Availability

The data supporting the findings of this study can be obtained from the corresponding author upon reasonable request. Access to patient data is governed by ethics and privacy considerations in accordance with institutional guidelines.

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
