# Peer review of "Treatment Outcomes in Spinal Tumors According to Patients’ Perspectives: A Focus on Indeterminate Spinal Instability"

_curroncol, 2025, doi:10.3390/curroncol32010038_

Round 1
Reviewer 1 Report
Comments and Suggestions for Authors
This is a very well-written and structured study of patients with metastatic spinal lesions. The SINS score is a useful tool to identify surgical patients but has a large range of "indeterminate" scores. The authors subdivided this indeterminate group to detect differences in higher vs lower scores and any relationship with need for surgery. Conclusion that indeterminate score does not predict need for surgery and that non-surgical treatments are effective for this population is useful for clinicians
Author Response
Thank you for your positive and encouraging feedback. We are pleased that you found our study well-written and structured. We appreciate your recognition of our efforts to subdivide the indeterminate SINS group and assess treatment outcomes. We are glad that our point about carefully basing surgical decisions on the indeterminate SINS score and the effectiveness of non-surgical treatments has been clearly conveyed and is useful for clinicians. Nevertheless, we made some revisions to the English to improve smoothness and readability throughout the manuscript.
Reviewer 2 Report
Comments and Suggestions for Authors
The authors report their experience with spinal metastatic tumors classified as indeterminate according to SINS classification. The presented cohort is small but the results specific. The dilemma of conservative vs surgical patients is always relevant and difficult to answer sometimes.
Why is "indeterminate" written twice in line 59?
It is a retrospective study (as you say) but your data is only from 2021, why?
You base the success of conservative treatment on self reporting of the patients in one single follow up? What about structural changes due to potential instability? Kyphosis or misalignment?
Please provide data on the follow up, how long was it? Mean follow up visits, complications that may happened etc.
You divided the 31 patients in conservative and surgical groups, you further subdivided these few patients according to their SINS score, you make your statistical analysis very weak.
I miss the exclusion criteria, the inclusion criteria are weak.
Did the patients of the conservative group get any other treatment (chemotherapy or radiation)?
In some parts of the text, the quality of English is poor.
I find your key point is important but you do not have the appropriate data to support it.
Author Response
The authors report their experience with spinal metastatic tumors classified as indeterminate according to SINS classification. The presented cohort is small but the results specific. The dilemma of conservative vs surgical patients is always relevant and difficult to answer sometimes.
Comment 1: Why is "indeterminate" written twice in line 59?
Response 1: Corrected /addressed
Comment 2: It is a retrospective study (as you say) but your data is only from 2021, why?
Response2: Thank you for your feedback. However, there seems to be a misunderstanding. As stated in the eligibility criteria, the data for this study were collected from patients attending our outpatient clinic with symptomatic or asymptomatic malignant spinal involvement between January 2021 and May 2023. This is clearly outlined in the manuscript, and we apologize for any confusion. We hope this clarifies the timeline of data collection for the study.
Comment 3: You base the success of conservative treatment on self reporting of the patients in one single follow up? What about structural changes due to potential instability? Kyphosis or misalignment?
Response 3: This is an important point. We do not base the success of treatment for tumor patients solely on imaging results, as the primary focus is on the patient's self-reported outcomes. It is important to emphasize that, in tumor patients, self-reported outcomes are often more significant than long-term radiological data. In clinical practice, a surgeon makes treatment decisions based not on imaging alone, but on the patient's preferences, wishes, and neurological status, especially when dealing with tumors. The key factor in evaluating success is the patient's reported well-being and functional status, rather than structural changes. Additionally, some patients chose to refuse imaging due to its perceived lack of impact on their treatment, which was considered a legitimate decision.
We also refer to the review by Balitsky AK et al., which highlights that in tumor patients, self-reported outcomes are most relevant. The discrepancy between clinician-reported and patient-reported outcomes suggests that accurate assessment of symptoms and consequent health-related quality of life (HRQoL) requires direct measurement from patients.
Balitsky AK, Rayner D, Britto J, et al. Patient-Reported Outcome Measures in Cancer Care: An Updated Systematic Review and Meta-Analysis. JAMA Netw Open. 2024;7(8):e2424793. doi:10.1001/jamanetworkopen.2024.24793
We agree that this important point was not properly addressed in our manuscript and may raise questions for the readers. To highlight this, we have added an explanatory paragraph in the discussion section to clarify our stance on the role of self-reported outcomes in tumor patient treatment success. This addition aims to emphasize that in tumor patients, self-reported outcomes often carry more weight than long-term radiological data, as they better reflect the patient's well-being and functional status rather than solely focusing on structural changes. Thank you very much for this comment!
“In tumor patients, treatment success is primarily based on self-reported outcomes, as these are often more relevant than long-term radiological data, reflecting the patient's well-being and functional status rather than structural changes.”
Comment 4: Please provide data on the follow up, how long was it? Mean follow up visits, complications that may happened etc.
Response 4: See results section for our changes /extension:
“Thirty-one patients (17 males [54.8%], 14 females [45.2%]) were included in the final analysis, collectively presenting with 147 highly suspected spinal neoplastic lesions. Patients were followed for a median of 133 days [IQR: 83.75, 283] and attended a median of 2.5 [IQR: 1,4] follow-up visits.”
Number of follow-up visits: median 2.5 [1,4]
Duration of follow-up: median 133 days [83.75, 283] -> approximately 4 months
This information was added to the first paragraph of the results section.
Comment 5: You divided the 31 patients in conservative and surgical groups, you further subdivided these few patients according to their SINS score, you make your statistical analysis very weak.
Response 5: The reviewer is correct that we do not have a high statistical power. However, comparing treatments was not the main scope of this study.
The aim of this retrospective study was to describe the treatment strategy by SINS scores. The intermediate SINS scores evaluated in this study were of particular interest since the treatment strategy is unclear for practitioners, and this information could lead to evidence-guided decisions for future patients.
Comment 6: I miss the exclusion criteria, the inclusion criteria are weak.
Response 6: We have extended the section on inclusion and exclusion criteria to provide further clarification.
Inclusion and Exclusion Criteria
Inclusion criteria
All patients aged 18 years or older who underwent at least one follow-up visit after treatment for spinal neoplastic lesions were considered eligible for inclusion in the study. We included patients only if both a standing X-ray and an MRI were available for review. Additionally, all patients had provided written informed consent, ensuring their voluntary participation in the study. Only patients with complete baseline clinical, radiological, and demographic data available for analysis were included, allowing for a comprehensive evaluation of treatment outcomes. Only patients without any relevant communication barriers were included to ensure effective communication.
Exclusion criteria
Patients were excluded if they met any of the following conditions: under 18 years of age, incomplete baseline clinical, radiological, or demographic data, no follow-up visit after treatment, absence of both standing X-ray and MRI, language barriers that impeded communication or informed consent, and pregnancy.
Comment 7: Did the patients of the conservative group get any other treatment (chemotherapy or radiation)?
Response 7: Thank you for your comment. As stated in the manuscript, within the intermediate group, 23 (79.3%) patients received radiotherapy, and 27 (93.1%) patients received chemotherapy. We apologize for any confusion, but this information is already included in the manuscript. (see Treatment Strategy of the Indeterminate Group)
Comment 8: In some parts of the text, the quality of English is poor.
Response 8: Thank you for your valuable feedback. We have revised the manuscript with the assistance of a native English speaker to improve the overall quality of the language. We focused on enhancing the clarity, flow, and smoothness of the text, addressing any areas where the language may have been unclear or awkward. We believe these revisions have strengthened the manuscript and improved its readability.
Comment 9: I find your key point is important but you do not have the appropriate data to support it.
Response 9: We understand your concern and acknowledge that the data supporting our key point may be limited. However, we believe that our findings, in conjunction with existing literature, provide a meaningful contribution to the discussion. While further research and more extensive data would be valuable, our study presents an important perspective based on the available evidence, which we hope adds to the ongoing dialogue in this area. We appreciate your input and will consider strengthening this aspect in future work.
Reviewer 3 Report
Comments and Suggestions for Authors
This retrospective study evaluates the clinical outcomes of patients with indeterminate SINS scores (7-12) for spinal metastases. By subdividing these scores into two groups, the research assesses treatment outcomes and validates decision-making strategies, providing evidence-based guidance for selecting between conservative management and surgical intervention in spinal metastatic disease. I have some comments:
1. The most urgent revision needed is in the methodology section, specifically regarding patient selection and standardization protocols. The criteria for selecting "the 10 most severe neoplastic lesions" must be explicitly defined with clear, reproducible parameters. Additionally, the standardization process for pain assessment and follow-up protocols should be thoroughly detailed to ensure methodological rigor and reproducibility of the study.
2. Secondly, the statistical analysis requires substantial enhancement. A power analysis should be included to justify the sample size, and the handling of multiple lesions per patient in the statistical analysis needs to be clearly explained. The addition of confidence intervals for main outcomes and appropriate multivariate analyses to account for potential confounding factors would significantly strengthen the study's validity.
3. The third critical revision pertains to the results presentation. The addition of a CONSORT-style flow diagram showing patient selection and follow-up would greatly improve clarity. Furthermore, Kaplan-Meier survival curves should be included to better illustrate outcomes over time, and the presentation of data from patients with multiple lesions needs to be more systematically organized and clearly displayed.
4. The fourth major revision should focus on strengthening the introduction section by including more recent systematic reviews and meta-analyses. The current knowledge gap needs to be more clearly articulated, and the rationale for subdividing SINS 7-12 into two groups should be more thoroughly supported with existing literature. This would better contextualize the study's significance within the current body of research.
5. Finally, the manuscript requires thorough language editing to improve clarity and consistency. Technical terminology should be standardized throughout the text, particularly regarding the use of "indeterminate" versus "intermediate." Complex sentences should be simplified, and tense usage should be consistent, especially in the methods section. This would enhance the overall readability and professional quality of the manuscript.
Comments on the Quality of English LanguageThe English could be improved to more clearly express the research.
Author Response
This retrospective study evaluates the clinical outcomes of patients with indeterminate SINS scores (7-12) for spinal metastases. By subdividing these scores into two groups, the research assesses treatment outcomes and validates decision-making strategies, providing evidence-based guidance for selecting between conservative management and surgical intervention in spinal metastatic disease. I have some comments:
Comment 1: The most urgent revision needed is in the methodology section, specifically regarding patient selection and standardization protocols. The criteria for selecting "the 10 most severe neoplastic lesions" must be explicitly defined with clear, reproducible parameters. Additionally, the standardization process for pain assessment and follow-up protocols should be thoroughly detailed to ensure methodological rigor and reproducibility of the study.
Response 2: Thank you for your valuable and constructive feedback. In response, we have revised the methodology section to clarify the patient selection process. Specifically, in cases with multiple lesions, we now explicitly describe that the most severe lesions were selected based on their SINS scores, with a maximum of 10 lesions per patient. Additionally, we have thoroughly revised the pain assessment and follow-up protocols to ensure methodological rigor and reproducibility. This revision was carried out with the expertise and support of our statistician, Dr. Nikki Rommers from the University of Basel's Statistics and Research Department. We greatly appreciate your detailed and supportive comments, which we believe have contributed to a better understanding of the study's methodology. Please refer to the whole method section for various additions and changes.
=> We added a section “pain evaluation” in methods part:
“For pain evaluation, we routinely use the Numeric Rating Scale (NRS) or Visual Analog Scale (VAS) during every consultation. Patients are asked to rate their pain, and if the reported number was lower, it was described as "better/improved." If the pain score remained the same, it was noted as "equal”, and if the score was higher, it was recorded as "worse." This standardized approach allowed for consistent and reliable tracking of pain levels throughout the study.”
For follow-up protocols, we suggested a follow-up period of 6 weeks for first FU and 12 weeks for 2nd FU (=primary outcome). However, due to the condition of the patients and the various medical appointments required for tumor care, the follow-up dates showed some variations. Despite these variations, we ensured that all patients received appropriate follow-up care within a reasonable timeframe.
Comment 2: Secondly, the statistical analysis requires substantial enhancement. A power analysis should be included to justify the sample size, and the handling of multiple lesions per patient in the statistical analysis needs to be clearly explained. The addition of confidence intervals for main outcomes and appropriate multivariate analyses to account for potential confounding factors would significantly strengthen the study's validity.
Response 2: We appreciate the reviewer’s suggestion regarding the power analysis. However, as this is a retrospective study, the sample size was determined by the available data, and no additional data can be obtained. Conducting a post hoc power analysis using the observed effect size would not provide meaningful insight, as it is generally not recommended for interpreting the validity or reliability of study findings (Hoenig & Heisey, 2001).
Reference: Hoenig, J. M., & Heisey, D. M. (2001). The Abuse of Power: The Pervasive Fallacy of Power Calculations for Data Analysis. The American Statistician, 55(1), 19–24. https://doi.org/10.1198/000313001300339897
Instead, we focused on reporting the observed confidence intervals and p-values to transparently convey the precision and statistical significance of our results. While we acknowledge the limitations of the sample size, the current study provides valuable insights and contributes to the existing body of knowledge.
We have also added the implications of the limited sample size in the limitations section and encourage future studies to confirm these findings with larger cohorts.
The multiple lesions per patient were accounted for in the statistical analysis by adding a random intercept per patient, which acknowledges that the lesions within the same patients are not completely independent of one another. We made this clearer in the statistical analysis section.
We agree with the reviewer that multivariate analysis would strengthen the study, however, with only very few patients treated conservatively, this was not possible with the available number of patients in this retrospective study.
Statistics
All analyses were performed using R version 4.3.1 (R Core Team, 2023). Patient characteristics, primary tumor characteristics, and the number of spinal neoplastic lesions were described by mean and standard deviation, median and interquartile range (IQR), or frequency and proportion, as appropriate. Additionally, scores of different components of the SINS score were described.
The success of the treatment was evaluated individually for metastasis and summarized by SINS category. The association between treatment and outcome was assessed via mixed-effects logistic regression models, including patient as a random intercept to account for the correlation between multiple lesions within a single patient. We present the odds ratio (OR) with 95% confidence interval (CI) for the fixed effect of treatment (surgery (yes over no) or radiotherapy (yes over no)).
Patient survival was described by the number and percentage of patients who died before data extraction and the median survival time after the date of first contact. The proportion of patients who died was stratified by treatment (surgery vs. no surgery), and a Kaplan-Meier curve was plotted to visualize survival by treatment group.
Also see:
Outcome of the Indeterminate Group
Within a single patient, all lesions showed consistent treatment outcomes. The majority of lesions in both the high and low indeterminate groups either improved (n = 59, 58.4%) or remained pain-free (n = 19, 18.8%). A total of 23 lesions (22.8%) showed no improvement after 90 days, but no patients experienced worsening symptoms (Table 5). The treatment outcome stratified by treatment type is displayed in Table 6.
The outcome in most patients with conservative treatment was good. 19 patients (65.5%) improved; 8 patients (27.5%) remained pain-free. The outcome in most patients undergoing surgery was good. Six patients (85,7%) improved, and one patient remained pain-free (14,3%).
The results of the models showed that there was no significant difference in treatment success between patient treated with and without surgery (OR 1.80 (95%-CI 0.00 to 700.61, p = 0.916)) or patient treated with and without radiotherapy (OR 1.05, (95%-CI 0.01 to 166.91, p = 0.986)). Also, within the group with intermediate SINS-score (SINS 7-12) no difference in treatment success was seen for patients with or without surgery or radiotherapy (OR 1.89 (95%-CI 0.00 to 68.43, p = 0.906) and OR 0.91, (95%-CI 0.00 to 223.75, p = 0.973), respectively).
Patient survival
Of all 31 patients, 53.6% patients died before the date of data extraction. The median survival time from the date of first contact in the patients who died was 214.5 days [IQR: 118.2, 319.5]. A total of 60.0% treated with surgery died, compared to 52.2% in the patients treated conservatively. Median survival time was 153 days [IQR: 102, 240] in the surgery group, and 223 days [IQR: 131, 317] in the conservative group.
Comment 3: The third critical revision pertains to the results presentation. The addition of a CONSORT-style flow diagram showing patient selection and follow-up would greatly improve clarity. Furthermore, Kaplan-Meier survival curves should be included to better illustrate outcomes over time, and the presentation of data from patients with multiple lesions needs to be more systematically organized and clearly displayed.
Response 3: We thank the reviewer for this comment. A patient flow chart has been added to the paper. This flow chart shows the number of records identified in the patients treated in the included clinic, after which we provide an overview of the number of lesions with intermediate SINS scores and surgical treatment. As we added both the number of patients and lesions to every section of the flow chart, this should provide the required insights in the data of patients with multiple lesions.
We added the Kaplan-Meier curve to the manuscript to illustrate survival over time. The figure is stratified by surgery.
We added a Flow Chart about patient selection process to the supplements.
Comment 4: The fourth major revision should focus on strengthening the introduction section by including more recent systematic reviews and meta-analyses. The current knowledge gap needs to be more clearly articulated, and the rationale for subdividing SINS 7-12 into two groups should be more thoroughly supported with existing literature. This would better contextualize the study's significance within the current body of research.
Response 4: Thank you for your valuable feedback. In response, we have added three additional studies from 2021, 2022, and 2024 in the second-to-last section of the introduction, addressing the topic of the grey zone (SINS 9-12). One study suggested a cutoff of 9, while another proposed a cutoff of 11. Based on these findings, we divided the group into higher and lower subgroups in an attempt to define a cutoff, although we ultimately did not establish one.
Comment 5: Finally, the manuscript requires thorough language editing to improve clarity and consistency. Technical terminology should be standardized throughout the text, particularly regarding the use of "indeterminate" versus "intermediate." Complex sentences should be simplified, and tense usage should be consistent, especially in the methods section. This would enhance the overall readability and professional quality of the manuscript.
Response 5: Thank you for your valuable feedback. We have revised the manuscript with the assistance of a native English speaker to improve the overall quality of the language. We focused on enhancing the clarity, flow, and smoothness of the text, addressing any areas where the language may have been unclear or awkward. We believe these revisions have strengthened the manuscript and improved its readability.
Round 2
Reviewer 2 Report
Comments and Suggestions for Authors
As I understand from the author's reply, they try to elucidate the treatment outcome in patients with SINS fracture 7-12 under the prism of the patient's anticipation.
According to the authors, most important is the self report of the patients the next few months than the radiological findings.
This is an interesting point of view but the manuscript is written in a misleading way, providing data from very few patients. The subjective outcome should be compared to the objective (prognosis, mortality/morbidity, radiological findings etc).
Furthermore, we cannot base our treatment decisions on short term results as it can cause irreversible damage to the patients.
I appreciate the changes you made to the text but you do not provide a comprehensive overview of treatments and treatment outcome as you mention in your introduction.
Try to rewrite the manuscript and change the title mentioning for example "treatments and treatment outcome accoring to patients´ perspective in cases with indeterminate instability". Longer term results are also necessary. The fact that they improved the next three or four months doesnt necessarily mean that the therapy was successful.
Author Response
2nd Revision Round: Author's Reply to the Review Report
General Comment Reviewer 2:
“As I understand from the author's reply, they try to elucidate the treatment outcome in patients with SINS fracture 7-12 under the prism of the patient's anticipation. According to the authors, the most important factor is the self-report of the patients in the next few months rather than the radiological findings.”
Response: The reviewer has provided a precise summary. We will include this point more explicitly in our manuscript to make it clearer for the reader. The reviewer is correct in suggesting that we should emphasize this more clearly to avoid any potential confusion.
“This is an interesting point of view, but the manuscript is written in a misleading way, providing data from very few patients. The subjective outcome should be compared to the objective (prognosis, mortality/morbidity, radiological findings, etc.).”
Response: This is an interesting point and could serve as a valuable approach for exploring this grey area, which has not yet been thoroughly addressed in the literature. However, our study is not able to provide this comparison. We will explicitly state this limitation in the manuscript and suggest it as a great idea for future research. Thank you for this valuable input!
“Furthermore, we cannot base our treatment decisions on short-term results as it can cause irreversible damage to the patients.”
Response: We absolutely agree! This is not the point our manuscript is trying to convey. There are established algorithms (SINS, NOMS) with clear thresholds, and we make sure to emphasize these algorithms and thresholds the manuscript (Introduction and Discussion). For example, if the SINS score falls above or below the "grey zone," clear treatment guidelines are available and align with our treatment strategies in clinical practice at our tertiary institution. Our focus is specifically on the grey zone, where hard data or clear thresholds do not apply. Furthermore, we did respect absolute surgery indications (or at least gave recommendation for surgery) if impending neurological damage was given. We are thankful for this comment, and we have revised the manuscript to clarify this point more explicitly.
“I appreciate the changes you made to the text but you do not provide a comprehensive overview of treatments and treatment outcome as you mention in your introduction.
Try to rewrite the manuscript and change the title mentioning for example "treatments and treatment outcome accoring to patients´ perspective in cases with indeterminate instability". Longer term results are also necessary. The fact that they improved the next three or four months doesnt necessarily mean that the therapy was successful.”
News title suggestion: "Treatment Outcomes in Spinal Tumors According to Patients' Perspectives: A Focus on Indeterminate Spinal Instability"
Response: We understand the reviewer’s concern that the manuscript should more clearly outline that our goal was to explore treatment strategies in the grey zone, rather than propose new cut-offs or guidelines. We have rewritten certain sections to clarify this point and to prevent any misinterpretation. While we cannot provide new long-term data, we want to highlight that survival in metastatic disease is generally poor, and long-term data is often not available due to the lack of follow-up and patient mortality. We have addressed the reviewer’s concerns by more clearly outlining the strengths and limitations of our data in the revised manuscript.
Changes in Manuscript 2nd Revision:
Title Treatment Outcomes in Spinal Tumors According to Patients’ Perspectives: A Focus on Indeterminate Spinal Instability
Abstract:
- Line 11: The objective of this study was to analyze treatment approaches and outcomes according to patients’ perspective for patients with indeterminate spinal instability …
- Line 17/18: No significant differences in self-reported outcomes were found between surgical and non-surgical treatments (p = 0.98, p = 0.18).
Introduction 53-58: Given the absence of clear thresholds in this grey zone, our effort was to emphasize patients' self-reported experiences over the following months as a key factor in evaluating outcomes in patients with indeterminate spinal instability, rather than relying solely on radiological findings. The aim of this study was to provide an overview of treatments and treatment outcomes, according to patients' perspectives, with a specific focus on those with indeterminate spinal instability.
Discussion:
Line 240: Among the outpatient population, we found that the majority of patients with indeterminate spinal instability can be managed non-surgically with favorable outcomes, according to patients' perspectives, regardless of whether their SINS scores fall on the higher or lower end of the indeterminate range.
In tumor patients, treatment success is primarily based on self-reported outcomes, which are often more relevant than long-term radiological data. These outcomes reflect the patient’s well-being and functional status rather than structural changes.
Line 300-308: To summarize, according to patients' self-reported outcomes, the majority of our patients facing an impending fracture risk due to malignancy either improved or remained stable with conservative treatment. From a patient's perspective, it may be suggested that non-surgical treatment can be pursued with greater confidence for individuals facing indeterminate spinal instability, given that there was no deterioration in neurological function and pain is manageable. Non-surgically treated patients with higher SINS scores within the indeterminate group exhibited similar self-reported outcomes to those with lower SINS scores.
Conclusion
Line 326-336: Most patients with impending spinal instability due to neoplastic lesions were treated non-surgically, yielding favorable outcomes according to patients’ perspective. The majority experienced improvements or remained asymptomatic or stable, underscoring the effectiveness of conservative management in these patients. Non-surgically treated patients with higher SINS scores within the indeterminate group exhibited similar outcomes to those with lower SINS scores.
According to patients' self-reported outcomes, conservative treatment for patients with indeterminate spinal instability due to neoplastic lesions was highly effective, with most patients showing improvement or demonstrating no progression of symptoms. Our findings support conservative management as a viable option for this patient population.
Reviewer 3 Report
Comments and Suggestions for Authors
The authors have responded to my comments well and revised their manuscript adequately. Thank you for your efforts.
Author Response
Thank you for your comments.
Round 3
Reviewer 2 Report
Comments and Suggestions for Authors
The new title helps the reader understand the real point of the manuscript.
Comments on the Quality of English LanguageAcceptable